# Diversity and Structure of Pelagic Zooplankton (Crustacea, Rotifera) in NE Poland

**Maciej Karpowicz** [1,*] and **Jolanta Ejsmont-Karabin** [2]

1 Department of Hydrobiology, Faculty of Biology, University of Białystok, Ciołkowskiego 1J, 15-245 Białystok, Poland

2 Research Station in Mikołajki, Nencki Institute of Experimental Biology, Polish Academy of Sciences, Pasteur 3, 02-093 Warsaw, Poland; j.karabin@nencki.gov.pl

* Correspondence: m.karpowicz@uwb.edu.pl; Tel.: +48-662-154-646

**Abstract:** This study presents the diversity and structure of pelagic zooplankton in north-eastern Poland. The research was conducted in 47 lakes with different trophic conditions in the middle of summer. Samples were collected close to the deepest part of the lakes to avoid the diverse benthic and littoral zones. We found 119 zooplankton species of which 32 were Cladocera, 16 were Cyclopoida, 4 were Calanoida, and 67 were Rotifera. We determined which species occurred most frequently in the region, as well as the species that were characteristic of different trophic conditions. We also recorded the presence of eight cold-adapted species which some of them are considered as glacial relicts (e.g., *Eurytemora lacustris*, *Heterocope appendiculata*, *Cyclops lacustris*). Our research revealed potential glacial refugia for planktonic species in 14 lakes of NE Poland. Our study suggests that the presence of stenotherm species may be an excellent indicator of the ecological status of deep lakes and could be considered in lake monitoring programs. Furthermore, we did not find *Bythotrephes longimanus* which has been reported from Poland. Instead, we found that *B. brevimanus* was the most common representative of the genus in the study area.

**Keywords:** species richness; plankton; community structure; trophy; glacial relicts; *Bythotrephes*

## 1. Introduction

Zooplankton are a key component of aquatic food webs that transfer energy and matter from primary producers to higher trophic levels and play a pivotal role in biogeochemical cycling [1]. Different groups of zooplankton are regulated by divergent environmental factors. For example, small-bodied species (rotifers and small cladocerans) are mainly regulated by "bottom-up" processes, while larger species are mostly regulated by "top-down" control by fish [2–4]. Furthermore, large-bodied zooplankters are more efficient at grazing on phytoplankton than their smaller competitors, which are restricted to consuming small particles [5–7]. Pelagic (open water or lacustrine) zooplankton species richness is important for ecosystem functioning, food web complexity, and ecosystem stability [8]. The pelagic zone is generally considered as a homogeneous habitat, however vertical environmental gradients (light, temperature, oxygen, food, predation pressure) create niches for different organisms [9]. The magnitude of these gradients increases with water transparency [10] which are the main factors promoting the diel vertical migration of zooplankton in clear lakes [11–13]. In less transparent lakes, food and oxygen conditions are often optimal in the surface waters, where visual predators are abundant and UV radiation levels are low. In contrast, clear lakes have high UV radiation at the surface water, and food resources are higher in deeper waters where visual predator abundance is often lower [10,14]. A deep chlorophyll layer formed by cryptophytes and diatoms is also a common phenomenon in clear lakes [15]. Cryptophytes which are very motile with mixotrophic feeding strategies [16] could benefit from higher bacteria biomass and

nutrient levels in the hypolimnion due to their low light needs [17]. Heavy and fast-sinking diatoms could benefit from greater water density [18–20] with higher nutrient and silicon concentration in the metalimnion [21–23]. This deep chlorophyll layer is a very important food source [23–27] and daytime refuge for large zooplankton to avoid visual predators [14,28–30]. The migration of large zooplankton to deeper waters creates favorable conditions for smaller species in the epilimnion [23]. Deep lakes that are well-oxygenated and have cold waters in the meta- and hypolimnion form a habitat for spring rotifer species [31] and stenotherm crustaceans which are sensitive to oxygen depletion [14]. Therefore, we assume that high diversity of pelagic zooplankton is linked to the good health of the lake ecosystem, with effective transfer of energy and matter in the food web.

Species distribution patterns are the cornerstone for biogeography, evolutionary biology, and ecology. The knowledge of how and why species are currently distributed in their geographical range are two fundamental questions in ecology and biogeography [32–34]. A literature review of lacustrine (pelagic) zooplankton by Dumont and Segers (1996) indicated that approximately 50 species of cladocerans are expected in lakes at any latitude, compared to approximately 150 rotifer species in temperate regions and 210 species in the tropical zone [35]. The above data were based on a large number of samples from few lakes which were multi-sampled from different sub-habitats. The other studies from a large number of lakes provided similar conclusions for a geographical region. Data on pelagic zooplankton in 1665 Canadian lakes revealed the presence of 83 crustacean species, but only 33 Cladocera [34]. Results from 2466 Norwegian lakes recorded the presence of 120 crustacean species and 77 Cladocera, but this study also included a major contribution from littoral crustaceans in its measurements of zooplankton species richness in the lakes [8]. Nevertheless, each Norwegian lake has an average of 14 species of microcrustaceans [8], and less than 10 species were reported from each Canadian lake [34]. Most studies on the large-scale geographic distribution of zooplankton mainly focus on the pelagic zones of lakes [34,36–38]. These results provide similar patterns in beta biodiversity (total number of species for a region) and indicate that pelagic zooplankton are quite homogeneous for each geographical region with 15–20 crustacean species that were found most frequently [34,36–38]. Results from three main Polish Lake Districts: Masurian (NE Poland), Pomeranian (NW Poland), and Great Poland (central Poland) indicated the presence of a similar set of zooplankton species, which were shaped by trophic conditions instead of geographical distance [36,39,40]. Zooplankton are widely recognized as an excellent indicator of lake trophic status [36,41–46]. Therefore, the pelagic zooplankton communities could be predictable based on the trophic status of lakes and the set of species for the region, while any deviation from the reference zooplankton community may indicate a disturbance in the food web and a deterioration of the ecological status of lakes.

This study presents the characteristics of pelagic zooplankton (Rotifera, Crustacea) in NE Poland. We distinguished a set of species that occurred most frequently in the region, as well as sets of species that were characteristic of different trophic conditions. We also present the differences in the zooplankton community structure (species richness, diversity, and biomass) in lakes with different trophic status. We hypothesize that the diversity of pelagic zooplankton should be related to the trophic status of lakes. A high diversity of pelagic zooplankton should promote an effective transfer of energy and matter in the food web and therefore indicate the health of the ecosystem.

## 2. Study Area and Methods

Our research was conducted in 47 lakes in Masurian Lakeland and Suwalki Lakeland (NE Poland) during the peak of summer stagnation in the years 2015–2019 (Table 1). Additionally, we distinguished four sampling stations on Lake Wigry (South Basin—14.1, Central Basin—14.2, North Basin—14.3, and Zadworze Bay—14.4) which are characterized by diverse morphometry and trophic conditions [9]. Some lakes were sampled twice (no. 14.2, 14.3, 14.4, 36, 38, 39), thus in total, we analyzed 56 plankton communities. The sampling stations in each lake were located close to the deepest point to avoid the diversified

benthic-littoral zone with macrophytes. Samples were collected using a 5-L Limnos sampler from the epilimnion, metalimnion, and hypolimnion of each lake. For zooplankton samples, ten liters of water from each of the three layers were individually filtered through a 50-μm plankton net and fixed with 4% formalin. The shallow lakes, without clear hypolimnion, were sampled from two layers. The total number of zooplankton samples that were analyzed was 148.

Field measurements included Secchi disc visibility (SDV), pH, temperature, and electrical conductivity using an HQ40D Multi Meter (Hach-Lange GmbH, Berlin, Germany). We also collected a water sample from the epilimnion in every lake for laboratory analysis of total phosphorus (TP), dissolved organic carbon (DOC), dissolved inorganic carbon (DIC), and chlorophyll-*a* (chl-*a*). These parameters were necessary for the calculation of trophic status. The analyses of TP were performed according to the molybdenate blue method [47]. The concentrations of DOC and DIC were analyzed via high-temperature catalytic combustion using a TOC-L Series (Shimadzu, Kyoto, Japan). The concentration of chl-*a* was analyzed using the spectrofluorometer FluoroProbe (bbe-Moldaenke, Germany) with Workstation. The trophic status of the harmonic lakes was calculated using the Carlson's trophic state index (TSI) as an average of three parameters: Secchi disc visibility (SDV), chl-*a*, and TP [48]. Lakes with a TSI below 40 were classified as oligotrophic, between 40–50 as mesotrophic, and above 50 as eutrophic. The state of dystrophy was evaluated using the hydrochemical dystrophy index (HDI) as an average of three equations, which include data for pH, electric conductivity, DIC, and DOC [49]. The HDI values between 50 and 65 indicate semi-dystrophic conditions, while values from 65 up to 100 indicate advanced dystrophy. Based on the above indexes, we distinguished 5 oligotrophic, 14 mesotrophic, 12 eutrophic, and 16 dystrophic lakes. The oligotrophic, mesotrophic, and eutrophic lakes are deep and large (Table 1) with the Secchi disc transparency 5.9 ± 1.4 m, 3.3 ± 1.4 m, and 1.4 ± 0. 6 m respectively. The dystrophic (humic) lakes are small, usually oval, without any outlets, and are surrounded by forest. Most of these lakes are shallow (Table 1) with sharp thermal and oxygen stratification, and anoxic conditions were observed from 1–2 m [14]. The dystrophic lakes were distinguished by high HDI values (Table 1) and also had other features typical of humic waters: yellow-brown color, acidic, a small quantity of mineral substances but a large amount of DOC [49].

Rotifers and crustaceans were identified to species and all individuals in the samples were enumerated. Ten length measurements were also made for each species and used to estimate the wet weight of crustaceans by applying the equation from Błędzki and Rybak (2016) [50]. The biomass of rotifers was established following the equation from Ejsmont-Karabin (1998) [51]. For the species richness and diversity analysis of zooplankton, we used averaged data from different layers for each lake. We investigated regional (NE Poland) and local species richness of pelagic zooplankton in different trophic conditions. We also compared diversity (number of species, Shannon index, Berger–Parker index) and community structures of crustaceans and rotifers in different trophic conditions.

**Table 1.** Morphometric and trophic characteristics of the studied lakes in north-eastern Poland. TSI—Carlson trophic state index; HDI—hydrochemical dystrophy index; SDV—Secchi disc visibility; oligo—oligotrophic; meso—mesotrophic; eu—eutrophic; dy—dystrophic. The lakes which were studied twice have been marked with an asterisk (*) after the lake number.

| no. | Lake Name | Date | Latitude (N) | Longitude (E) | Surface (ha) | Max Depth (m) | Trophic Status | TSI | HDI | SDV (m) |
|---|---|---|---|---|---|---|---|---|---|---|
| 1 | Białe Filipowskie | 29 July 2019 | 54°11′56″ | 22°38′59″ | 132.4 | 52.0 | oligo | 29.6 | 35.0 | 7.5 |
| 2 | Gaładuś | 29 July 2019 | 54°10′31″ | 23°25′22″ | 728.6 | 54.8 | oligo | 33.8 | 35.3 | 5.7 |
| 3 | Serwy | 29 July 2019 | 53°54′06″ | 23°12′15″ | 460.3 | 41.5 | oligo | 38.5 | 35.0 | 3.8 |
| 4 | Jegocin | 23 July 2019 | 53°39′52″ | 21°41′54″ | 127.4 | 36.1 | oligo | 39.7 | 37.7 | 7.0 |
| 5 | Leleskie | 24 July 2019 | 53°38′37″ | 20°50′38″ | 423.5 | 49.5 | oligo | 39.7 | 37.3 | 5.5 |
| 6 | Jaczno | 22 July 2015 | 54°16′48″ | 22°52′25″ | 41.0 | 19.0 | meso | 40.5 | 29.1 | 2.9 |
| 7 | Hańcza | 24 July 2015 | 54°15′48″ | 22°48′35″ | 311.4 | 108.5 | meso | 40.8 | 31.1 | 4.2 |
| 8 | Buwełno | 22 July 2019 | 53°52′48″ | 21°51′37″ | 360.3 | 49.1 | meso | 42.4 | 30.7 | 2.4 |
| 9 | Majcz Wielki | 31 July 2019 | 53°46′49″ | 21°27′14″ | 163.5 | 16.4 | meso | 42.8 | 34.1 | 2.7 |
| 10 | Kuc | 30 July 2019 | 53°49′12″ | 21°24′23″ | 98.8 | 28.0 | meso | 43.5 | 34.7 | 4.0 |
| 11 | Białe Wigierskie | 16 July 2018 | 54°01′53″ | 23°05′26″ | 100.2 | 34.0 | meso | 44.3 | 33.3 | 5.7 |
| 12 | Busznica | 19 July 2018 | 53°56′38″ | 23°05′00″ | 49.4 | 48.0 | meso | 44.3 | 34.8 | 6.6 |
| 13 | Szurpiły | 24 July 2015 | 54°13′43″ | 22°53′51″ | 89.0 | 46.8 | meso | 45.8 | 31.3 | 2.9 |
| 14.1 | Wigry, South Basin | 18 July 2018 | 54°00′54″ | 23°03′38″ | 2118.3 | 74.2 | meso | 45.4 | 30.9 | 5.2 |
| 14.2 | Wigry, Central Basin | 7August 2015 | 54°02′53″ | 23°05′40″ | 2118.3 | 74.2 | meso | 48.3 | 36.6 | 2.0 |
| 14.2 * | Wigry, Central Basin | 26 July 2016 | 54°02′53″ | 23°05′40″ | 2124.3 | 74.2 | meso | 45.9 | 32.3 | 3.2 |
| 14.3 | Wigry, North Basin | 7 August 2015 | 54°03′50″ | 23°04′50″ | 2118.3 | 74.2 | meso | 48.5 | 36.6 | 2.2 |
| 14.3 * | Wigry, North Basin | 26 July 2016 | 54°03′50″ | 23°04′50″ | 2118.3 | 74.2 | meso | 46.9 | 31.5 | 3.5 |
| 14.4 | Wigry Zadworze Bay | 7 August 2015 | 54°04′21″ | 23°05′09″ | 2118.3 | 74.2 | meso | 47.6 | 36.1 | 2.6 |
| 14.4 * | Wigry Zadworze Bay | 26 July 2016 | 54°04′21″ | 23°05′09″ | 2118.3 | 74.2 | meso | 44.8 | 33.2 | 4.9 |
| 15 | Probarskie | 30 July 2019 | 53°49′26″ | 21°22′40″ | 201.4 | 31.0 | meso | 46.3 | 33.6 | 3.8 |
| 16 | Brzozolasek | 1 August 2019 | 53°36′58″ | 21°44′14″ | 155.9 | 17.2 | meso | 49.4 | 36.0 | 1.2 |
| 17 | Kalwa | 24 July 2019 | 53°38′36″ | 20°45′27″ | 562.2 | 31.7 | meso | 49.6 | 34.8 | 2.0 |
| 18 | Okrągłe | 16 July 2018 | 54°01′14″ | 23°01′21″ | 10.7 | 4.6 | meso | 49.7 | 31.4 | 3.1 |
| 19 | Mikołajskie | 31 July 2019 | 53°47′22″ | 21°34′56″ | 497.9 | 25.9 | meso | 49.9 | 34.8 | 2.0 |
| 20 | Jagodne | 22 July 2019 | 53°55′19″ | 21°42′33″ | 942.7 | 37.4 | eu | 50.3 | 29.6 | 2.2 |
| 21 | Boczne | 31 July 2019 | 53°57′40″ | 21°44′46″ | 183.3 | 17.0 | eu | 51.1 | 32.2 | 1.4 |
| 22 | Wiartel | 1 August 2019 | 53°36′04″ | 21°41′49″ | 178.6 | 29.0 | eu | 51.7 | 37.7 | 1.6 |
| 23 | Ryńskie | 31 July 2019 | 53°54′43″ | 21°29′40″ | 670.8 | 50.8 | eu | 52.5 | 34.3 | 1.0 |
| 24 | Kierźlińskie | 24 July 2019 | 53°48′03″ | 20°44′32″ | 92.8 | 44.5 | eu | 53.1 | 32.7 | 1.5 |
| 25 | Długie Wigierskie | 18 July 2018 | 54°01′33″ | 23°01′23″ | 80.0 | 14.8 | eu | 53.5 | 30.3 | 1.6 |

**Table 1.** *Cont.*

| no. | Lake Name | Date | Latitude (N) | Longitude (E) | Surface (ha) | Max Depth (m) | Trophic Status | TSI | HDI | SDV (m) |
|---|---|---|---|---|---|---|---|---|---|---|
| 26 | Leszczewek | 19 July 2018 | 54°04′21″ | 23°03′47″ | 21.0 | 6.5 | eu | 54.4 | 32.1 | 2.3 |
| 27 | Nidzkie | 23 July 2019 | 53°37′46″ | 21°32′38″ | 1818.0 | 23.7 | eu | 55.4 | 35.4 | 0.9 |
| 28 | Garbaś | 24 July 2018 | 54°08′05″ | 22°37′20″ | 140.6 | 48.0 | eu | 58.0 | 30.1 | 2.0 |
| 29 | Necko | 24 July 2018 | 53°51′47″ | 22°57′50″ | 400.0 | 25.0 | eu | 58.9 | 32.2 | 1.1 |
| 30 | Miłkowskie | 22 July 2019 | 53°56′31″ | 21°52′14″ | 23.7 | 15.0 | eu | 59.4 | 32.3 | 0.5 |
| 31 | Juno | 31 July 2019 | 53°53′38″ | 21°17′47″ | 380.7 | 33 | eu | 62.9 | 32.7 | 0.7 |
| 32 | Widne | 28 July 2016 | 54°00′44″ | 23°07′25″ | 1.9 | 4.0 | dy | 57.5 | 58.6 | 1.4 |
| 33 | Klimunt | 26 July 2019 | 53°42′23″ | 21°26′55″ | 12.8 | 4.0 | dy | 63.8 | 60.4 | 0.3 |
| 34 | Wesołek | 25 July 2019 | 53°35′33″ | 21°30′44″ | 7.0 | 3.0 | dy | 50.7 | 66.9 | 1.2 |
| 35 | Zdrużno | 26 July 2019 | 53°38′13″ | 21°20′59″ | 6.8 | 5.0 | dy | 51.2 | 69.8 | 1.8 |
| 36 | Suchar Wielki | 28 July 2016 | 54°01′40″ | 23°03′20″ | 11.0 | 9.6 | dy | 50.9 | 67.1 | 2.5 |
| 36 * | Suchar Wielki | 18 July 2018 | 54°01′40″ | 23°03′20″ | 11.0 | 9.6 | dy | 56.7 | 68.2 | 2.2 |
| 37 | Suchar I | 19 July 2018 | 54°05′07″ | 23°00′54″ | 1.3 | 2.0 | dy | 62.0 | 66.5 | 1.1 |
| 38 | Suchar II | 28 July 2016 | 54°05′14″ | 23°01′03″ | 2.6 | 9.5 | dy | 53.7 | 69.4 | 1.9 |
| 38 * | Suchar II | 19 July 2018 | 54°05′14″ | 23°01′03″ | 2.6 | 9.5 | dy | 58.1 | 71.1 | 1.5 |
| 39 | Wądołek | 28 July 2016 | 54°06′39″ | 23°02′38″ | 1.2 | 15.0 | dy | 70.0 | 68.6 | 1.2 |
| 39 * | Wądołek | 24 July 2018 | 54°06′39″ | 23°02′38″ | 1.2 | 15.0 | dy | 62.9 | 70.0 | 1.1 |
| 40 | Dembowskich | 28 July 2016 | 54°02′18″ | 23°03′33″ | 3.1 | 3.5 | dy | 55.2 | 74.5 | 2.5 |
| 41 | Sęczek | 23 July 2019 | 53°43′41″ | 21°32′47″ | 3.8 | 3.5 | dy | 56.0 | 72.2 | 0.9 |
| 42 | Gryżlewskie | 23 July 2019 | 53°43′26″ | 21°33′03″ | 4.3 | 5.0 | dy | 50.7 | 71.4 | 1.8 |
| 43 | Borkowskie | 23 July 2019 | 53°43′16″ | 21°32′57″ | 2.9 | 5.0 | dy | 53.7 | 73.2 | 1.2 |
| 44 | Kruczy Staw | 25 July 2019 | 53°39′41″ | 21°24′21″ | 2.1 | 8.0 | dy | 47.7 | 73.3 | 2.0 |
| 45 | Kruczek | 25 July 2019 | 53°39′36″ | 21°24′08″ | 4.2 | 4.0 | dy | 51.3 | 74.2 | 1.5 |
| 46 | Kruczek Mały | 25 July 2019 | 53°39′28″ | 21°25′01″ | 2.6 | 9.0 | dy | 46.7 | 74.6 | 1.7 |
| 47 | Konopniak | 29 July 2019 | 53°35′07″ | 21°33′09″ | 9.5 | 4.0 | dy | 50.4 | 79.9 | 1.2 |

We presented the full list of crustacean and rotifers species as a Pareto chart to identify the most important species. Species frequencies in the Pareto chart are represented in descending order by bars, and the cumulative total of the sample is represented by the curved line where 80% distinguished the most important species. An estimate of total species richness—Chao2 [35,52,53], which calculates the estimated true species diversity of a sample, was applied to determine the likely number of species that could be identified in each of the studied lakes. Linear regressions and ANOVA were conducted to describe the relationship between the number of samples (=lakes) and the number of species calculated with procedures that included 10 randomizations of sampling order using the freeware program EstimateS, version 7.52 [54]. We used one-way ANOVA with Fisher's F tests to determine the effect of trophic status on diversity and biomass of crustacean and rotifers communities. One-way ANOVA was followed by Tukey's HSD (honestly significantly different) to test all pairwise differences between means, which were marked on the box plots. Canonical correspondence analysis (CCA) ordinations were used to present the community structure of zooplankton in different trophic conditions. Statistical analyses were performed with XLSTAT Ecology (Addinsoft, New York, NY, USA).

## 3. Results

A total of 52 crustacean species (32 Cladocera, 16 Cyclopoida, 4 Calanoida) and 67 rotifer species were found in the pelagic zone of the 47 lakes in NE Poland (Figure 1, Table S1). There was relatively large diversity in the genus *Daphnia* (6 species) and *Cyclops* (5 species). The genus *Bosmina* was also diverse despite the presence of only 3 species, because there were 5 morphs of *Bosmina (Eubosmina) coregoni* (Table S1). We also found the presence of species that could be considered glacial relicts: *Eurytemora lacustris* (lake no. 1, 2, 3, 7, 11, 13, 14, 28), *Heterocope appendiculata* (no. 1, 3, 4, 5, 10, 11), *Cyclops abyssorum* (no. 14), *Cyclops lacustris* (no. 2, 4, 11), *Holopedium gibberum* (no. 38), *Daphnia longiremis* (no. 14), *Bythotrephes brevimanus* (no. 1, 3, 5, 8, 21), and *Bythotrephes* cf. *lilljeborgi* (no. 2, 7, 14). We did not find *Bythotrephes longimanus*, which was the only species within this genus that was previously reported from Poland. Instead, our results indicated that *B. brevimanus* is the most common representative of this genus in NE Poland. However, the far east of NE Poland appears to be a glacial refugia for *B.* cf. *lilljeborgi*. Both of these crustacean species are newly reported in the Polish fauna. The vast majority of rotifers consisted of commonly occurring and often dominant species except for three rotifer species found in dystrophic lakes. These include *Trichocerca simoneae*, which recently invading Polish lakes [55], *Ploesma tricanthum* which is rare in north-eastern Poland, and rare *Brachionus sessilis* was observed in the metalimnion of mesotrophic lake no. 12. In many cases, pelagic communities were enriched with single individuals brought into open waters from the littoral zone, mainly by *Lecane*, which is very abundant among macrophytes.

The species composition of planktonic crustaceans was relatively similar in different lakes. The most frequent species (euconstant, above 75%) were *Diaphanosoma brachyurum*, *Mesocyclops leuckarti*, and *Daphnia cucullata* (Figure 1A). According to the Pareto chart (Figure 1A), twelve additional species were an important component of the crustacean communities (*Thermocyclops ointhonoides*, *Daphnia longispina*, *Bosmina longirostris*, *Ceriodaphnia quadrangula*, *Leptodora kindtii*, *Eudiaptomus gracilis*, *Bosmina coregoni*, *Bosmina crassicornis*, *Chydorus sphaericus*, *Eudiaptomus graciloides*, *Daphnia cristata*, and *Cyclops scutifer*) (Figure 1A). Among other less frequent crustaceans, there were species with high (or specific) habitat requirements or littoral species (Figure 1A, Table S1). The most frequent rotifer species (above 75%) were *Keratella cochlearis*, *Polyarthra vulgaris*, *Asplanchna priodonta*, and *Polyarthra remata* (Figure 1B). However, in the case of Rotifera, 19 additional species were also an important component of the community (Figure 1B).

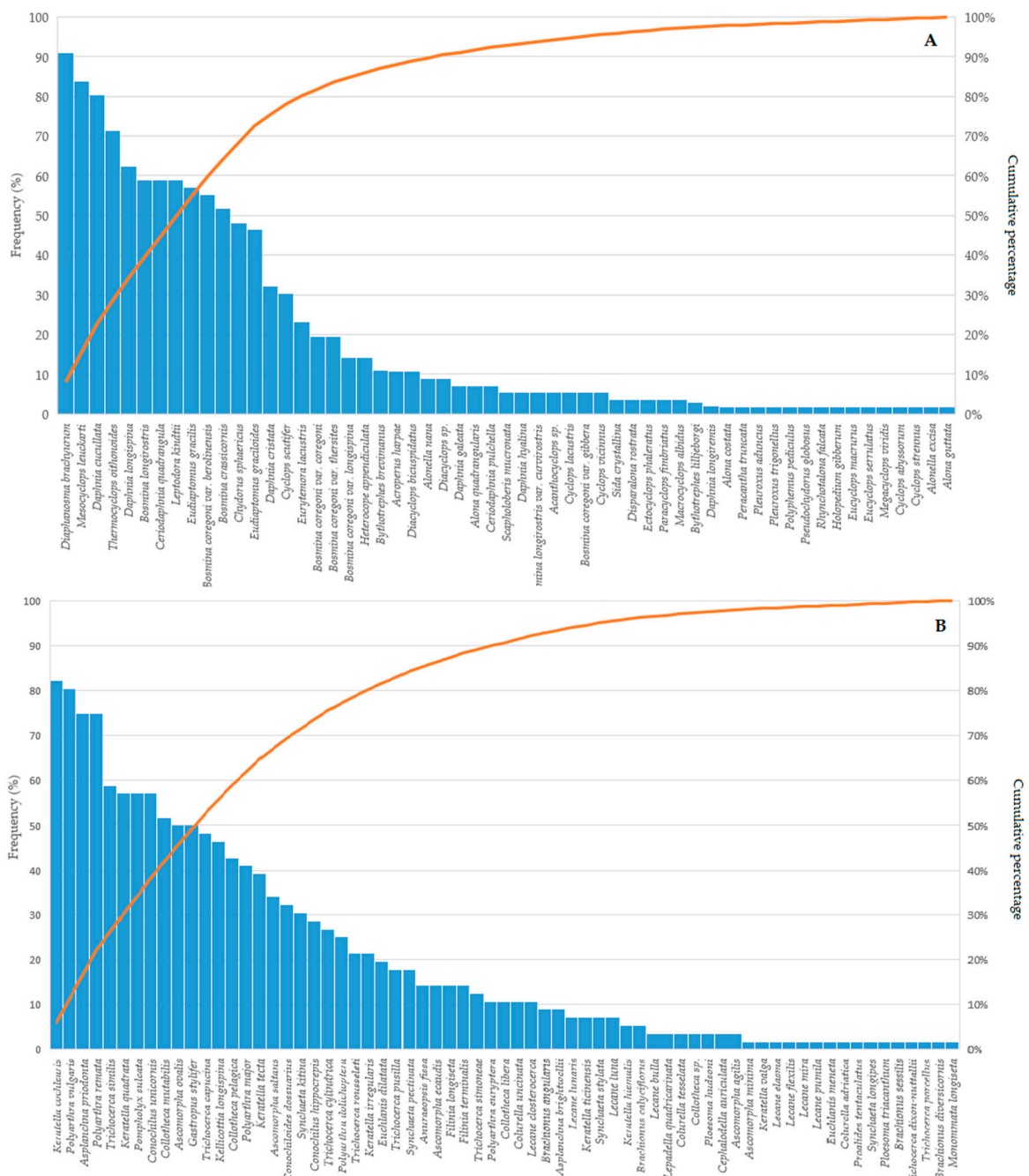

**Figure 1.** The frequency of Crustacea (**A**) and Rotifera species (**B**) in lakes of north-eastern (NE) Poland presented as Pareto charts. Both charts present a full list of species.

There were significant differences in the number of crustacean (F = 53.95; *p* < 0.0001) and rotifer species (F = 13.04; *p* < 0.0001) in different trophic conditions. The highest number of crustacean species was found in oligotrophic conditions, where we can expect 15–20 species in every lake (Figure 2A). The lowest number of crustacean species was found in the dystrophic conditions where we can expect 5–7 species in every lake (Figure 2A). The highest number of rotifer species was found in mesotrophic and eutrophic lakes, while the lowest was found in dystrophic lakes (Figure 2B). The trophic conditions also influence the diversity of crustaceans (F = 21.94; *p* < 0.0001) and rotifers (F = 10.86; *p* < 0.0001). The lowest values of the Shannon index for crustaceans and rotifers were in dystrophic lakes (Figure 2C,D), with an average of 1.23 ± 0.34 and 0.86 ± 0.32, respectively. There were

no significant differences in the Shannon index of eutrophic, mesotrophic, and oligotrophic lakes (Figure 2C,D), where the average Shannon index for crustaceans and rotifers was $2.05 \pm 0.34$ and $1.47 \pm 0.51$, respectively. There was also one eutrophic lake (no. 27) with a very high diversity of planktonic Crustacea and Rotifera (Figure 2C,D). The Berger–Parker dominance index for crustacean and rotifers was the highest in dystrophic lakes while it was significantly lower in eutrophic, mesotrophic, and oligotrophic lakes (Figure 2E,F).

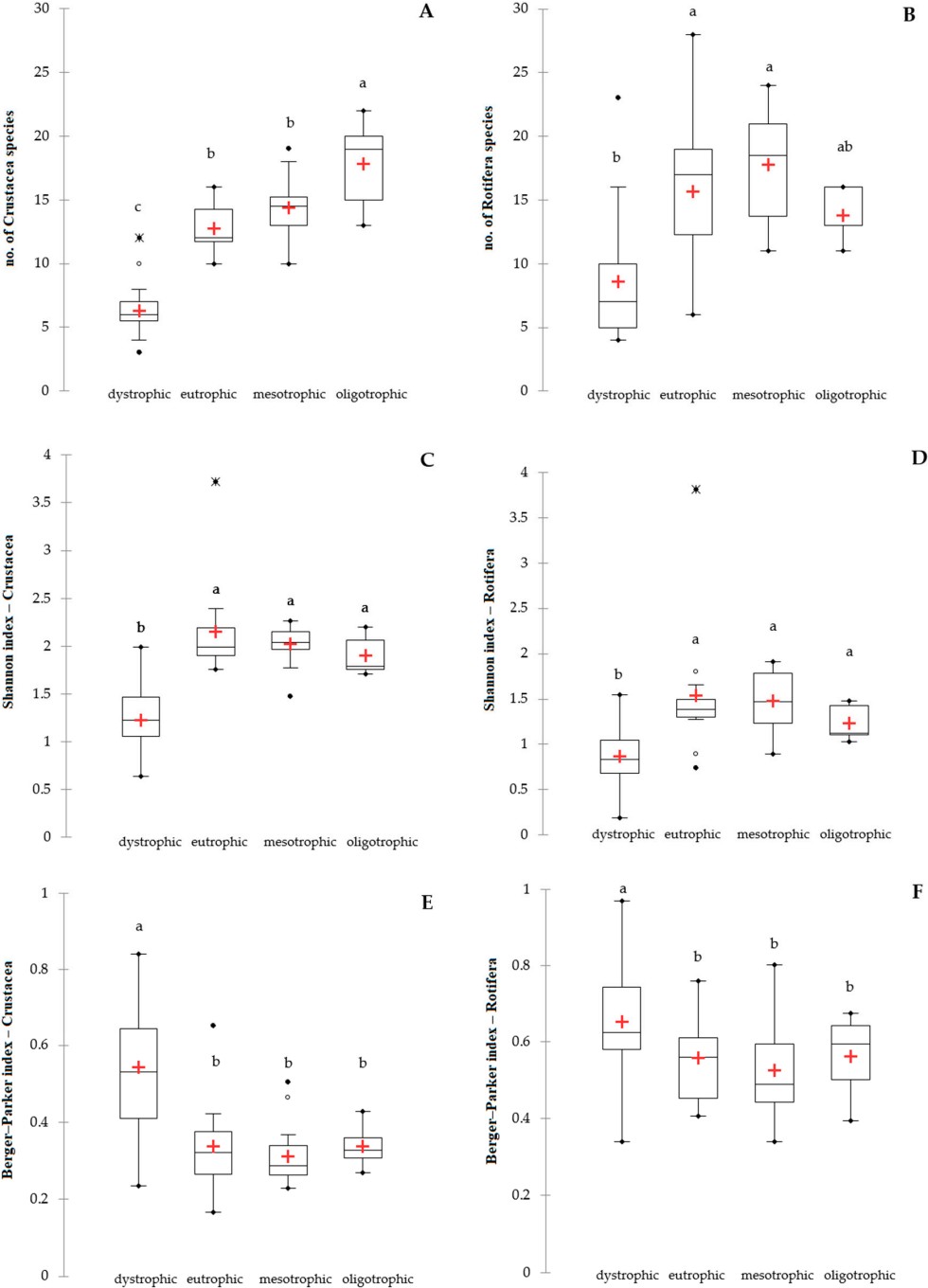

**Figure 2.** Diversity of crustacean (**A**,**C**,**E**) and rotifer (**B**,**D**,**F**) communities in different trophic conditions presented as an average number of species (**A**,**B**), Shannon index (**C**,**D**) and Berger–Parker dominance index (**E**,**F**). The lower and upper limits of the boxes are the first and third quartiles, respectively. The crosses correspond to the means and the central horizontal bars are the medians. Points above or below the whiskers are the outliers and extreme values. The different letters (a, b, c) above the box plots denote significantly different values at $p < 0.05$; the same letters denote no statistically significant differences.

The analysis of the relationship between the number of samples and the number of species showed no differences in low and high trophy harmonic lakes (Figure 3), while the number of crustacean species was strongly related with the trophic status and markedly lower in eutrophic lakes. Both numbers of rotifer and crustacean species were the lowest in dystrophic lakes (Figure 3). These results are in marked contrast to the true number of species obtained by the application of the Chao2 estimator. It suggests 48 crustacean and 40 rotifer species in harmonic lakes, thus only slightly more than those found in our studies, but in dystrophic lakes this difference was much higher. Chao2 estimation gives a value of 82 rotifer species, thus nearly twice that recorded in our research (i.e., 43). The results of Chao2 estimation for crustaceans were surprising. Unlike rotifers, crustaceans were the poorest taxonomically (40 species) in dystrophic lakes. A surprisingly high number of rotifer species (103!) was recorded in eutrophic lakes. It resulted from the high number of singletons (12) and only one double record.

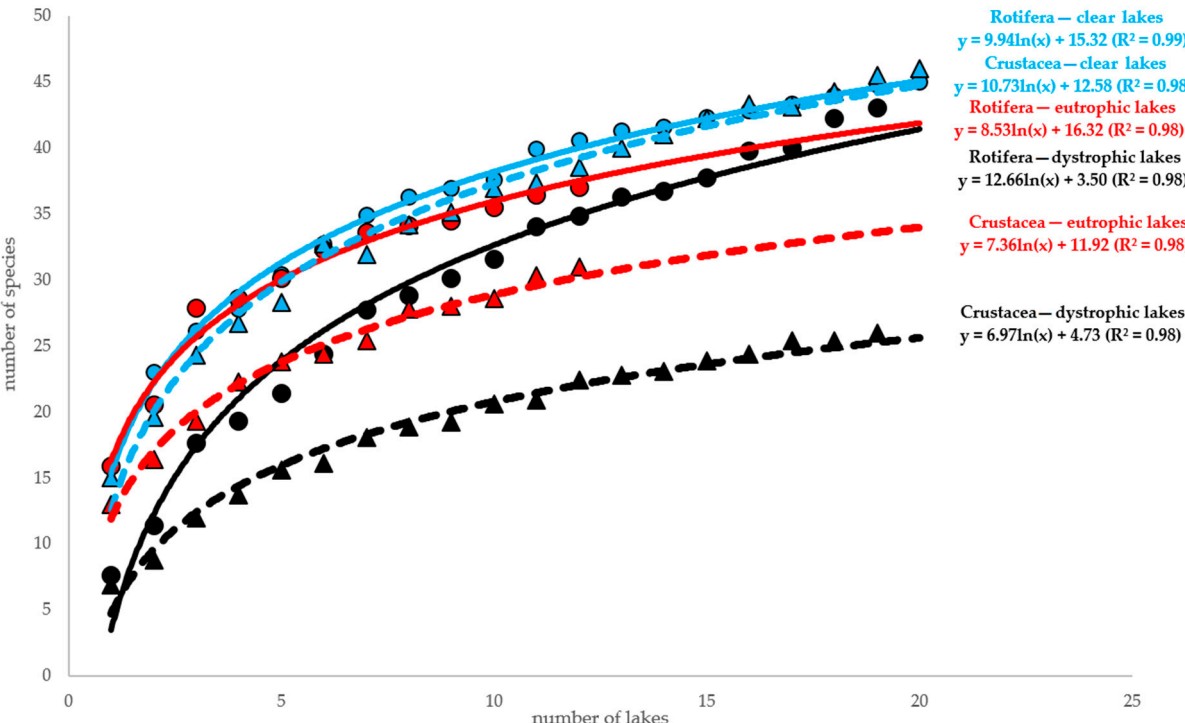

**Figure 3.** The relationship between the number of samples (=lakes) and the number of species calculated with 10 randomizations of sampling order. Clear lakes represent oligotrophic and mesotrophic conditions; circles represent Rotifera, triangles—Crustacea.

There were significant differences in the biomass of crustacean zooplankton in different trophic conditions (F = 53.95; *p* < 0.0001). The highest biomass of Crustacea was found in mesotrophic and eutrophic lakes, with an average of $2.84 \pm 1.65$ mg L$^{-1}$, and lower in oligotrophic and dystrophic lakes, with an average of $1.20 \pm 0.91$ mg L$^{-1}$ (Figure 4A). There were no significant differences in Rotifera biomass in different trophic conditions (F = 1.23; *p* = 0.31), and the average biomass of rotifers was $0.35 \pm 0.52$ mg L$^{-1}$ (Figure 4B). However, there were a few outliers in rotifer biomass (Figure 4B), which was caused by the mass development of *Asplanchna priodonta* (and *A. brightwellii* in lake no. 19) to level as high as 40.57 mg L$^{-1}$ in the metalimnion of lake no. 42.

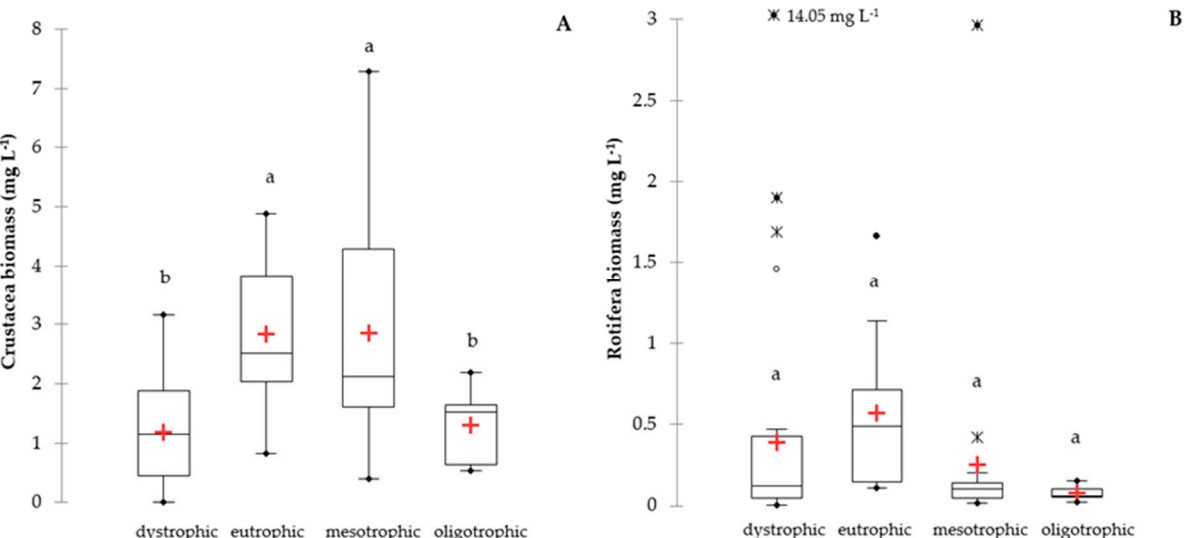

**Figure 4.** Average biomass of crustacean (**A**) and rotifers (**B**) from vertical profiles of lakes with different trophic status. Descriptive statistics as in Figure 2.

There were large differences in zooplankton community composition between dystrophic and the remaining lakes (Figure 5 A–C). The dystrophic lakes were distinguished by the dominance of Rotifera and small cladocerans in the zooplankton communities (Figure 5A). The rotifer communities in dystrophic lakes were dominated by *Asplanchna priodonta* and *Trichocerca similis* (Figure 5B), while crustacean communities were dominated by *Ceriodaphnia quadrangula* and *Eudiaptomus gracilis* (Figure 5C). There was a large list of species that were found only in dystrophic lakes, among them was *Holopedium gibberum*, *Polyphemus pediculus*, *Scapholeberis mucronata*, *Ascomorpha minima*, *Brachionus diversicornis*, *Colurella tesselata*, *Keratella valga*, *Lecane elasma*, *Lecane mira*, *Ploesoma triacanthum*, *Synchaeta longipes*, and *Trichocerca simoneae* (Table S1). Several of these species are rare for the Polish and European fauna.

Further differences in the species structure of zooplankton communities between low trophy (oligotrophic and mesotrophic) and eutrophic lakes were observed. The low trophy lakes were distinguished by the presence of stenotherm crustaceans and the higher relative abundance of *Bosmina coregoni* var. *coregoni*, *Bosmina crassicornis*, and *Daphnia* species (Figure 5C). The rotifer communities in those lakes had a higher relative abundance of *Conochilus* spp., *Collotheca pelagica*, *Synchaeta pectinata*, *Synchaeta kitina*, *Filinia terminalis*, *Kellicottia longispina*, *Ascomorpha ovalis*, and *Polyarthra vulgaris* (Figure 5B). The crustacean communities in the eutrophic lakes had a higher relative abundance of *Bosmina coregoni* var. *thersites*, *Bosmina coregoni* var. *berolinensis*, *Chydorus sphaericus*, and *Eudiaptomus graciloides* (Figure 5B). The rotifer communities in eutrophic lakes had a higher share of *Trichocerca pusilla*, *Trichocerca rousseleti*, *Conochiloides dossuarius*, *Collotheca mutabilis*, *Pompholyx sulcata*, *Polyarthra major*, *Polyarthra remata*, and *Trichocerca cylindrica* (Figure 5B).

There was a large difference in the crustacean communities in the vertical profiles of clear lakes (Figure 5D), whereas rotifer were relatively uniformly distributed throughout the water column. The deeper water layer had a higher relative abundance of *Daphnia longispina*, *Daphnia hyalina*, *Daphnia cristata*, *Bosmina longirostris*, *Bosmina coregoni* var. *berolinensis*, *Bosmina coregoni* var. *thersites*, *Bosmina coregoni* var. *gibbera*, *Bosmina longirostris*, *Cyclops* spp., and species considered as glacial relicts (except *H. appendiculata*). The epilimnion communities of clear lakes were dominated by *Daphnia cucullata*, *Diaphanosma brachyurum*, *Bosmina crassicornis*, *Leptodora kindtii*, and *Chydorus sphaericus* (Figure 5D).

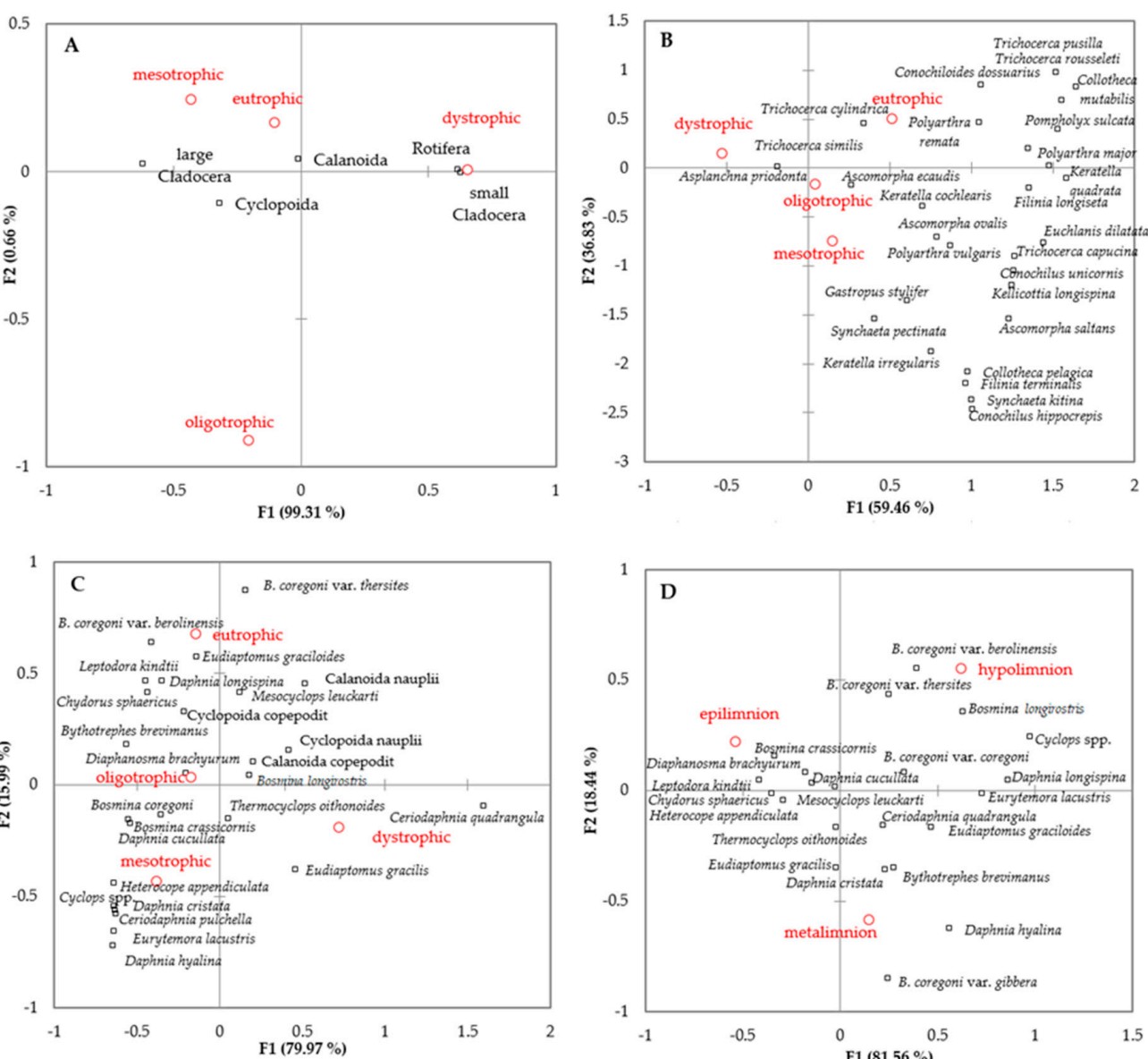

**Figure 5.** Zooplankton communities structure in different trophic conditions. Major zooplankton groups (**A**), Rotifera (**B**) and Crustacea (**C**) communities in different trophic conditions. The vertical distribution of crustacean zooplankton in clear lakes (**D**).

## 4. Discussion

The results of our study present characteristics of lacustrine zooplankton in NE Poland. Most studies on zooplankton diversity are focused on the pelagic zone. However, the presence of typical littoral species creates a problem in the assessment of richness, due to the overestimation of species richness because only a small part of the littoral species pool exploits the pelagic habitat [8]. Therefore, our study focused on the more typical planktonic species and sampling was carried out close to the deepest part of the lakes to avoid the diverse benthic-littoral zone with macrophytes. We have found 119 zooplankton species (67 Rotifera, 32 Cladocera, 16 Cyclopoida, 4 Calanoida) in the pelagic zone of the 47 lakes in north-eastern Poland. Higher species richness, 151 species (86 Rotifera, 41 Cladocera, 20 Cyclopoida, 4 Calanoida), was previously recorded in 79 lakes in NW Poland [39]. Nevertheless, the list of pelagic species was almost the same in lakes of NW and NE Poland [39], and the higher species richness in NW Poland was related to the inclusion of typical littoral

species and differs in terms of taxa importance (e.g., *Bosmina* species). We distinguished the common lacustrine species (>70%) in Polish lakes, which included *Keratella cochlearis*, *Polyarthra vulgaris*, *Asplanchna priodonta*, *Mesocyclops leuckarti*, *Daphnia cucullata*, *Diaphanosoma brachyurum*, and *Thermocyclops oithonoides*. These species were also very numerous and often dominated the zooplankton biomass. Furthermore, we distinguished other species that are important and a common component of lacustrine zooplankton in Poland.

Our results pointed to the relatively large diversity of the genus *Daphnia*, *Bosmina,* and *Cyclops* in northern Poland. The *Daphnia* genus contained six species (Figure 1) and the most common was *D. cucullata,* which often dominated in the epilimnion, while the other *Daphnia* species dominated in the deeper water layer. The cladoceran *D. cucullata* is widely distributed and very frequently reported from the whole of Europe and often dominates in lakes where fish predation is intense, due to a completely colorless body [50,56–58]. Contrasting patterns of body size for coexisting *Daphnia* species that segregate by habitat is a well-known phenomenon [59] because they have different life-history strategies, habitat preferences, behavior, and vulnerability to predation [2,6,60,61]. The coexisting of 3–5 *Daphnia* species at each station of Lake Wigry has been known for a hundred years [9,62,63].

The genus *Bosmina* in NE Poland contains three species, but *Bosmina coregoni* include 5 morphs which were previously regarded as distinct species. Recent evidence based on morphological characters and molecular phylogenies indicated that this is one quite variable species [64,65]. We found that *B. coregoni* morphs '*thersites*' and '*longispina*' prefer eutrophic lakes, while the morph '*berolinensis*' prefer low trophic lakes. Beyond the typical form of the common *Bosmina longirostris*, we also have found the morph '*curvirostris*' in dystrophic lakes.

The pelagic Cyclopoida of NE Poland are characterized by the very high frequency of two species (*Mesocyclops leuckarti*, *Thermocyclops oithonoides*) which often dominate the zooplankton biomass and the large diversity of the genus *Cyclops*. The cold-adapted *Cyclops* contains five species in our study (Table S1), which was the same as in NW Poland [39]. Nonetheless, this genus is widespread in the Palearctic and contains about 30 species of which at least 11 occur in Central Europe [66–68]. Thus, our results are rather underestimated and we could expect a higher diversity of *Cyclops* genus in northern Poland, which requires further study.

The Calanoida of NE Poland were represented by the two common species (*Eudiaptomus gracilis* and *E. graciloides*) and two stenotherms regarded as "glacial relicts" (*Eurytemora lacustris* and *Heterocope appendiculata*). We did not find the other large calanoid *Limnocalanus macrurus*, which occurred here in the second half of the 20th century [36,40]. The other potential glacial relicts in NE Poland were *Cyclops lacustris*, *Daphnia longiremis*, and *Bythotrephes* spp. We did not find *Bythotrephes longimanus* which was reported from Poland; instead, we found that *B. brevimanus* is the most common representative of the genus in NE Poland. We also reported the presence of *B.* cf. *lilljeborgi* in three lakes in the far east of NE Poland. Both of these species are new records for the Polish fauna. The confusion in this genus could be a result of treating the genus *Bythotrephes* as monotypic represented by only one quite variable species (*B. longimanus*); however, the last detailed revision revealed at least seven species [69–71]. The geographical distribution of the genus *Bythotrephes* by Litvinchuk and Litvinchuk [69] and Korovchinsky [70] also indicated that *B. brevimanus* is the most common in Central Europe, while *B. longimanus* is known from the Alps and northern boreal zone [70]. Finally, our research revealed potential glacial refugia for planktonic species in 14 lakes of NE Poland, and most of them were not known before. This indicated that glacial relicts may be more common in deep lakes in northern Poland than previously thought. These species almost disappeared from Polish lakes in the eighties, as a result of accelerated eutrophication [9], and currently, a greater relative abundance may indicate improvement of the ecological status of deep lakes in northern Poland.

Taking into account that small organisms tend to have a cosmopolitan distribution [72,73] and reproduction strategies of rotifers are strongly oriented towards achieving maximum dispersal [74,75], many rotiferologists assume that rotifers are to a large extent

cosmopolitan, and ecological barriers, rather than geographical, are decisive in their distribution [76–78]. Nevertheless, endemics are observed among rotifers, especially in the ancient Lake Baikal [79], as an affection of the time where probably completely due to the genetic drift which produces diversification within each population possibly without any dependence from the variability of conditions and habitats [80]. It is therefore not unusual that in relatively young and poorly isolated lakes of north-eastern Poland, we do not observe endemic species [40] and there are also no rotifer relicts. Summer communities of Rotifera in harmonic lakes were built of common and cosmopolitan species belonging to the genera *Keratella* (*K. cochlearis* and *K. quadrata*), *Polyarthra* (*P. major*, *P. remata,* and *P. vulgaris*), *Pompholyx sulcata,* and others. The same has been concluded based on research conducted in four lake systems of north-eastern Poland [40].

The high diversity of pelagic zooplankton is associated with greater temporal stability in species composition [38] and could be a response to the good health of the lake ecosystem with effective transfer of energy and matter in the food web. Thermal stratification separates the epilimnion from the hypolimnion and thus increases the potential for the coexistence of species with different habitat requirements [23,81], especially for species that require cold and well-oxygenated water [82,83]. The vertical differentiation of environmental conditions in lakes also reduces competition between different groups of zooplankton and allows the utilization of food particles of different sizes [11]. Therefore, the low species richness of zooplankton may indicate a large disturbance in the lake ecosystem because zooplankton richness declines considerably with the increasing eutrophication process [42,45]. However, our results indicated that only crustacean species richness was related to trophic status with the highest number of species in oligotrophic lakes due to the presence of stenotherm species in the deeper water layer. We also found a large difference in the crustacean community in vertical profiles of clear lakes, where larger species prefer deeper water layer and smaller dominated in the epilimnion. Thus, the migration of large zooplankton to deeper waters creates favorable conditions for smaller species in the epilimnion [23]. The Rotifera species richness in our study was not related to the trophic status and we found a similar number of species (average and estimated) in different trophic conditions. The diversity of rotifers and crustaceans were similar in eutrophic, mesotrophic, and oligotrophic conditions. This conclusion confirms Karabin's (1985) observations made for 64 lakes in north-eastern Poland. He stated that pelagic rotifer communities were composed of 10 to 17 species in more than 80% of the studied lakes, and that the number of species was not related to the trophic state of lakes [84]. Thus, our results pointed out that the diversity of pelagic zooplankton alone is not the best indicator of the ecological status of lakes. However, the zooplankton community structure is commonly used as an indicator of the eutrophication process [41–44,84,85]. It may be concluded that although the number of species is not dependent on the trophy of lakes, the community structure, i.e., participation of different species in the community is strongly related to trophic status. The low trophy lakes were characterized by a higher relative abundance of large Cladocera, while highly eutrophic lakes had a higher relative abundance of Rotifera and small Cladocera. The larger cladocerans are superior competitors for resources, but they are more vulnerable to visual predation by fish [86]. In clear lakes, large species migrate to dark waters during the day and return to food-rich epilimnetic waters at night to access phytoplankton [87,88]. Therefore, if the deeper water layer provides sufficient refuge for large species during the day where it could persist under strong fish pressure [89]. In this study, we also pointed out that the presence of stenothermic species with high environmental requirements indicated good health of deep lakes because these species are very sensitive to environmental deterioration. Furthermore, we distinguished the species characteristics for low and high trophic status.

Only the dystrophic lakes are clearly distinguished by the lower diversity and species richness of crustaceans and rotifers. The number of zooplankton species found in dystrophic lakes was about half of the diversity in the other lakes, and the community structure was commonly dominated by one or two species (*Asplanchna priodonta*, *Ceriodaphnia quadrangula*, and *Eudiaptomus gracilis*). In the case of *A. priodonta,* the mass development reached

even 40.57 mg L$^{-1}$. Factors that limit the development of large Cladocera in dystrophic (humic) lakes, despite low fish pressure and a large amount of food resources are still discussed [14]. Among the factors limiting the development of zooplankton are humic stress connected to the high concentrations of humic substances and DOC [90,91], low food quality [92], low pH [49], UV radiation [93,94], sharp temperature, and oxygen gradients from the surface [95]. We suggest that zooplankton in humic lakes are strongly limited by all of the above factors and only a few species could thrive under such conditions. However, dystrophic lakes have a large number of unique species that were found only there, and among them there were also rare species like *Holopedium gibberum* and *Ploesoma triacanthum*. The cladoceran *H. gibberum* is widely distributed in water bodies of the boreal zone while in southern latitudes it is restricted mostly to relict lakes with soft water, poor in dissolved salts of mainly calcium and magnesium [96].

### 5. Conclusions

The zooplankton communities in open water zones of different lakes in one geographical region (NE Poland) are quite homogeneous and consist of a similar set of species. There were four euconstant species of Crustacea (*Diaphanosoma brachyurum*, *Mesocyclops leuckarti*, *Daphnia cucullata*, and *Thermocyclops ointhonoides*) and Rotifera (*Keratella cochlearis*, *Polyarthra vulgaris*, *Asplanchna priodonta*, *Polyarthra remata*). We distinguished twelve more crustacean species and nineteen more rotifer species that were important components of the zooplankton communities in NE Poland. Despite the presence of quite similar species, we revealed some differences in zooplankton communities that were related to different trophic conditions. Crustacean species richness increased with decreasing trophic status, while rotifer species richness was not related to trophic status. We also distinguished rotifer and crustacean species whose abundance was related to different trophic conditions. Thus, changes in zooplankton communities seem to be a good indicator of the eutrophication process. We also pointed out that the presence of planktonic glacial relicts with high environmental requirements is an excellent indicator of the good ecological status of deep lakes.

**Supplementary Materials:** The following are available online at https://www.mdpi.com/2073-4441/13/4/456/s1, Table S1: Frequency (%) of zooplankton taxa occurrence in lakes of NE Poland.

**Author Contributions:** M.K.: writing—original draft preparation, conceptualization, field analyses and sampling, analyzed the crustacean zooplankton, performed statistical analysis, visualization; J.E.-K.: analyzed the Rotifera zooplankton, statistical analysis, writing—review and editing. All authors have read and agreed to the published version of the manuscript.

**Funding:** This research was supported by the Polish National Science Centre by grant number 2017/01/X/NZ8/01151 and 2016/21/B/NZ8/00434.

**Institutional Review Board Statement:** Not applicable.

**Informed Consent Statement:** Not applicable.

**Data Availability Statement:** The data presented in this study are the part of this article and are available in the supplementary materials.

**Acknowledgments:** The authors are thankful to Joanna Kozłowska for her help in the collection of samples and to Adam Więcko and Helena Samsonowicz for their assistance in the water chemistry analyses.

**Conflicts of Interest:** The authors declare no conflict of interest.

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
