# Peer review of "Diversity and Structure of Pelagic Zooplankton (Crustacea, Rotifera) in NE Poland"

_water, doi:10.3390/w13040456_

Round 1
Reviewer 1 Report
Dear Editor and authors
article highlighted importance od zooplankton research as platform for many other studies and topics. There are few issues to solved (clean to clear, lower layer to deeper layer, binomial nomenclature rules in Tabl S1), but, generally article could be accepted after minor revision. My specific comments you can find through the attaced ms.

Reviewer 2 Report
The manuscript written by Maciej Karpowicz et al. describes the diversity and the structure of summer pelagic zooplankton communities in lakes in NE Poland. They pointed out the high similarity of species composition of pelagic zooplankton, but the dystrophic lake can be characterised as simpler zooplankton communities. They also showed that Rotifers are more diverse group than crustacean zooplankton, especially in dystrophic environment. They also presented that the zooplankton community of low trophic lakes can be distinct from the others by the presence of stenotherm species. Their research revealed that lakes in NE Poland are regarded as glacial refugia for planktonic species.
The Introduction is written clearly and the hypotheses are reasonable, Methods, however, should be improved. The authors collected an extensive dataset, 47 lakes were sampled. However, some of them were sampled twice. Since the zooplankton communities exhibit temporal heterogeneity, the authors must give information about the yearly differences in zooplankton communities of the lakes. They used diversity indices to describe the complexity/entropy of zooplankton communities. The Berger-Parker index refers to the proportion of the most dominant species in the community. It increases with the domination of the most common species, hence, it is highly biased by sample size and richness. Moreover, the index tells nothing about the available information about the community. Therefore, it is rarely used by ecologists. Instead, true diversity or the effective number of species are used for comparing communities. The data is suitable to investigate community changes on local and regional scales by diversity partitioning.
The authors used one-way ANOVA to test the effect of trophic status on diversity and biomass distribution of zooplankton groups. I am not sure if the authors tested the assumptions of ANOVA. The authors handled Rotifers and Crustaceans separately, please describe why.
All my comments can be found in the attached pdf file.I recommend the manuscript for publication with “minor revisions”.

Reviewer 3 Report
I found the manuscript
Diversity and structure of pelagic zooplankton (Crustacea, Rotifera) in NE Poland
interesting and stimulating.
probably for this reason (stimulation) I ask for a more extended discussion. and a consultation of the recent published papers of Water (there are some regarding the same field).
All my suggestion are directly indicated in the pdf file.
Thank you for the attention.

Round 2
Reviewer 3 Report
Dear Authors,
I, as a referee, but also as a member of the editorial board of Water, already asked a more accurate use of references and/or already done studies. At present I cannot state that this has been accepted.
For example, a huge quantity of data on lakes (small and large) are available from all the World, reporting the species composition, and these list, in the order of importance (= species number), Rotifera, Cladocera, Cyclopoida, Calanoida.
You, however, cite repeatedly the only (although important, in lakes number) research where such an order is (for micro-crustaceans) not confirmed.
I suppose that you have a reason for this consideration, and a scientific explanation usable for your data, too. But this does not appear, and the reader ask himself: Why such a large diversity of results (those proposed and those compared with)? If this research rises problems of comparison (the two results differ in terms of taxa importance) and you do not discuss this point, the paper becomes someway deficient.
In addition, also the comparison with the historical study of Dumont and Segers 1996 is not appropriate. They studied a set of 11 lakes, large and multisampled (over many years), hence (also in this case) not comparable with your methodological approach. but in this case you discuss and find the solution: Dumont and Segers find more species per lake because they used data coming from many years and researches.
You have researches comparable with your data: a Norway survey on more than 2,000 lakes, and/or a Poland survey on species richness from 75 lakes. I suggest to limit your attention to these researches more than those which weaken the discussion.
always in the perspective of keeping a (scientific) position, I suggest to better study the available literature (not only data papers, but also opinion papers) even in the Water journal. It is interesting, from this point of view, that a scientific research as the present one, does not find any references on the journal that has to host it (are we sure that Water is the right journal for this study?). I'm sure, also to justify your choice of the journal, and my presence here in the Editorial board, that Copepoda, Freshwater zooplankton, species richness and distribution, and related keywords, will allow you to find material (and useful suggestion) even in Water, to improve the criticized parts of introduction and discussion.
finally, I suggest to avoid critical citations (if any solution can be proposed in the discussion) and substitute them with newly available data to reinforce (and not to weaken) your results (see directly on the pdf file these evidenced parts)
This suggestion is (more or less) the same of the previous review, and if You prefer to not accept such a suggestion, please carefully explain your motivation.
Thank you for the attention,
Genuario Belmonte

Round 3
Reviewer 3 Report
Dear Authors,
I have to apologize for my unclear (evidently) suggestion regarding the reference to the paper on 1665 Canadian lakes: That paper lists zooplankton taxa in order of importance, and that order of importance (referred to the species number) is embarrassingly different from our "European" data. Thus my suggestion was NOT related to the position of taxa in the list (hence suggesting to respect the same order list) but to the importance of taxa. In that paper Calanoida are the most important taxon, in the rest of literature Calanoida are the last taxon. This paper rises problems of interpretation and is not useful in discussion of your data (Polish). For this reason I suggested to avoid any reference to the number of Calanoida.
The comparison of methods, however, is the point to be carefully considered (multisampled habitats? large-small?, pelagic-neritic, and so on) and I appreciate your effort to specify your goal in comparison with other studies.
As regarding the cited PhD thesis of Rahkola-Sorsa (2008) I have to attract your attention on the fact that she refers to the abundance of individuals (dominance), when she speaks about the Calanoida importance, and not to a prevalence of Calanoida (only 6!) on other Crustacea, in terms of species number (the total number of Crustacea species was 36). Thus also the cited paper does not agree with the Canadian data. In North Europe large lakes Calanoida dominate zooplankton in terms of specimen abundance. In Canadian lakes Calanoida are the most abundant Crustacea in terms of species number (!). The two papers rely upon different topics.
In any cases, I like your work and your data, and I would like to develop also with you a discussion on interpretation of species richness data and/or on geographic distribution of biodiversity. This is the reason why I find interesting that your numbers do not completely agree with the area/species rule coming from McArthur and Wilson theory (do you know the work of Ebert-Balko, 1987?). I already tried to propose an alternative motivation for the species richness of lake Bajkal (Belmonte 2012) hence I'm happy if this discussion goes on.
Thank you for the stimulation,
and Hope to meet you again in the future.
small corrections are ri-proposed in the text (see annexed file) and you can progress with the completion of the final version of the manuscript, without sending again to me the corrections.
Regards,
Genuario Belmonte

Author Response
Dear Professor Genuario Belmonte,
Thank you very much for your work with this manuscript. All corrections in the text have been applied, and we include in the discussion your alternative motivation for the species richness of Lake Baikal.
Yours sincerely,
Maciej Karpowicz